# Clinical Characteristics and Outcomes of Patients with Primary and Secondary Myelofibrosis According to the Genomic Classification Using Targeted Next-Generation Sequencing

**DOI:** 10.3390/cancers15153904

**Published:** 2023-07-31

**Authors:** Marta Garrote, Mónica López-Guerra, Eduardo Arellano-Rodrigo, María Rozman, Sara Carbonell, Francesca Guijarro, Marta Santaliestra, Ana Triguero, Dolors Colomer, Francisco Cervantes, Alberto Álvarez-Larrán

**Affiliations:** 1Hematopathology Section, Pathology Department, Hospital Clínic Barcelona-IDIBAPS, 08036 Barcelona, Spain; lopez5@clinic.cat (M.L.-G.); mrozman@clinic.cat (M.R.); fguijarro@clinic.cat (F.G.); dcolomer@clinic.cat (D.C.); 2Centro de Investigación Biomédica en Red de Cáncer (CIBERONC), Instituto de Salud Carlos III, 28029 Madrid, Spain; 3Hematology Department, Hospital Clínic Barcelona-IDIBAPS, 08036 Barcelona, Spain; arellano@clinic.cat (E.A.-R.); scarbonell@clinic.cat (S.C.); atriguero@clinic.cat (A.T.); fcervan@clinic.cat (F.C.); aalvar@clinic.cat (A.Á.-L.); 4Hematology Department, Hospital Universitari Mutua Terrassa, 08221 Terrassa, Spain; msantaliestra@mutuaterrassa.cat; 5Medicine Campus, Faculty of Medicine and Health Sciences, Universitat de Barcelona, 08036 Barcelona, Spain

**Keywords:** myelofibrosis, myeloproliferative neoplasms, genomic classification, prognosis, personalized medicine

## Abstract

**Simple Summary:**

Myelofibrosis is a heterogeneous disease regarding its mutation landscape as well as its clinical presentation and outcome. A genomic classification with prognostic implications of myeloproliferative neoplasms has been previously proposed, however, this classification has hardly been implemented in myelofibrosis. The aim of our work is to evaluate this genomic classification in a large series of myelofibrosis patients from a single institution, taking into account whether cases are primary or secondary. We found that both primary and secondary myelofibrosis have distinctive molecular landscapes and that genomic profiling provides accurate information regarding prognosis and disease progression.

**Abstract:**

Myelofibrosis (MF) is a heterogeneous disease regarding its mutational landscape, clinical presentation, and outcomes. The aim of our work is to evaluate the genomic classification of MF considering whether it is primary or secondary. One-hundred seventy-five patients, 81 with primary MF (PMF) and 94 with secondary MF (SMF) were hierarchically allocated into eight molecular groups. We found that *TP53* disruption/aneuploidy (*n* = 16, 9%) was more frequent (12% versus 6%) and showed higher allele burden (57% versus 15%, *p* = 0.01) in SMF than in PMF, and was associated with shorter survival (median 3.5 years). Mutations in chromatin/spliceosome genes (*n* = 72, 41%) represented the most frequent genomic group in PMF. Homozygous *JAK2* mutation (*n* = 40, 23%) was enriched with old patients with SMF after long-standing polycythemia vera, whereas MF with heterozygous *JAK2* mutation (*n* = 22, 13%) was similarly distributed among PMF and SMF. MF with *CALR* mutation (*n* = 19, 11%) predominated in post-essential thrombocythemia MF. The remaining genomic groups were infrequent. *TP53* disruption, chromatin/spliceosome mutation, and homozygous *JAK2* mutation were associated with significantly shorter survival and higher risk of progression. In conclusion, genomic classification reveals different pathogenic pathways between PMF and SMF and provides relevant information regarding disease phenotype and outcomes.

## 1. Introduction

Myelofibrosis (MF) is a heterogeneous myeloproliferative neoplasm (MPN), from both a clinical and a prognostic point of view [1]. Thus, while some patients remain asymptomatic and do not require treatment for years, others suffer from a debilitating and progressive disease in which constitutional symptoms, transfusion dependence, and splenomegaly impair quality of life and result in reduced survival [1,2]. The disease can appear de novo (primary MF, PMF) or as a result of disease progression from another MPN (secondary MF, SMF), such as polycythemia vera (PV) or essential thrombocythemia (ET) [1,3]. Although there are no differences in terms of clinical manifestations and treatment, both types of MF are usually considered different entities [1,3,4].

Clinical variability reflects the complex and heterogeneous landscape of MF. Most patients harbor a driver mutation in signaling recurrent genes, such as *JAK2*, *CALR*, or *MPL*. These mutations are mutually exclusive and do show correlation with disease phenotype and prognosis [5,6,7]. Additional mutations in genes involved in epigenetic regulation, affecting the RAS signaling pathway or the tumor suppressor *TP53* also play a fundamental role in the transformation of the malignant cell and, therefore, in its clinical expression [8,9,10,11].

Recently, a molecular classification of MPN including eight genomic groups has been proposed as a useful tool for personalizing the prognosis and treatment of MPN patients [10].

It is a hierarchical classification based on an algorithm that contains eight categories; molecular and chromosomal abnormalities associated with worse prognosis prevail in the classification. Cases presenting with *TP53* disruption or chromatin/spliceosome mutations are included in categories one and two, respectively. Categories three to six include MPN with driver mutations (*CALR*, *MPL*, or *JAK2*) that lack definitory abnormalities within categories one and two. Patients with *JAK2* mutations are separated into two categories, according to allele burden (categories five and six). Finally, categories seven and eight include cases with no known mutations and MPN with other driver mutations, respectively.

This molecular classification offers relevant prognostic information in addition to the MPN driver mutation and it allows us to identify those patients at higher risk of progression, disease complications, and death, regardless of the disease phenotype. However, it has hardly been implemented in clinical practice. The aim of the present study is to evaluate the genomic classification of MF in a large series of MF patients from a single institution, taking into account whether cases are primary or secondary.

## 2. Materials and Methods

All patients diagnosed with PMF or SMF at our institution with available DNA samples were included in the present study. The diagnosis was carried out according to World Health Organization 2016 criteria [12]. IPSS and MYSEC prognostic scores (see Appendix A) at diagnosis were calculated for all patients as previously described [13,14]. Informed consent was obtained and the study was approved by the Ethics Committee of Hospital Clínic (Barcelona, Spain).

DNA samples were isolated from peripheral blood at diagnosis (*n* = 131, 75%) or during follow-up (*n* = 44, 25%) and stored in the Hematopathology Collection (Biobank Hospital Clínic—IDIBAPS; R121004-094). DNA samples that were collected during follow-up were obtained shortly after diagnosis or at a time in which clinical characteristics have not changed significantly since diagnosis. Targeted sequencing by Next-Generation Sequencing (NGS) was performed using commercial panels from Sophia Genetics (customized Myeloid Solution panel MIC_v1, MiSeq system, Illumina platform) or Thermo Fisher Scientific (Oncomine Myeloid Assay, Ion GeneStudio S5 system, Ion Torrent platform). Myeloid Solution by Sophia genetics included the following 32 genes: *ABL1, ASXL1, BRAF, CALR, CBL, CEBPA, CSF3R, CSNK1A1, DNMT3A, ETV6, EZH2, FLT3, HRAS, IDH1, IDH2, JAK2, KIT, KMT2A, KRAS, MPL, NPM1, NRAS, PTPN11, RUNX1, SETBP1, SF3B1, SRSF2, TET2, TP53, U2AF1, WT1, ZRSR2.* Oncomine Myeloid Research Assay by Thermo Fisher contained the following genes: *ABL1, ASXL1, BCOR, BRAF, CALR, CBL, CEBPA, CSF3R, DNMT3A, ETV6, EZH2, FLT3, GATA2, HRAS, IDH1, IDH2, IKZF1, JAK2, KIT, KRAS, MPL, MYD88, NF1, NPM1, NRAS, PHF6, PRPF8, PTPN11, RB1, RUNX1, SETBP1, SF3B1, SH2B3, SRSF2, STAG2, TET2, TP53, U2AF1, WT1, ZRSR2.* GRCh37 (hg19) was used as the reference genome in both panels. Variant calling and analysis were carried out using the SOPHiA DDM™ platform and Ion Reporter Software 5.18™. Further information comparing each NGS panel is detailed in Appendix A.

Only variants with a variant allele frequency (VAF) ≥ 1% were considered and classified within the following categories: pathogenic, likely pathogenic, uncertain significance, likely benign, and benign. Variant categorization was carried out using genomic databases and predictors for the general population and cancer (COSMIC, ClinVar, Seshat, Franklin, and Varsome) and published references in myeloid neoplasms [10,15,16]. Only pathogenic and likely pathogenic variants were taken into account for further analysis. Pathogenic and likely pathogenic variants found with their VAF can be found in Appendix A.

The genomic classification was performed as reported by Grinfeld et al. [10]. Briefly, patients were hierarchically allocated into eight molecular categories: MF with *TP53* disruption or aneuploidy (*TP53* mutation, Chr17pLOH or Chr5-/Chr5q-); MF with ≥1 genetic aberration in chromatin or spliceosome genes (*EZH2, IDH1, IDH2, ASXL1, PHF6, CUX1, ZRSR2, SRSF2, U2AF1, KRAS, NRAS, GNAS, CBL,* Chr7/7qLOH, Chr4q/LOH, *RUNX1, STAG2,* and *BCOR*); MF with *CALR* mutation; MF with *MPL* mutation; MF with homozygous *JAK2* mutation; MF with heterozygous *JAK2* mutation; MF with other driver mutation; MF with no known mutation. For genomic allocation, cytogenetic data from medical records when available, and mutational and copy number variation information derived from NGS were employed.

Qualitative variables were compared through a Chi-squared test and quantitative variables through Student’s *t*-test, Mann-Whitney U test, or ANOVA, as appropriate. Overall survival and time-to-event curves were drawn using the method of Kaplan-Meier with a log-rank test for comparisons. Multivariate analyses of the factors predicting the different outcomes were carried out by Cox regression. A Cox regression model with a competing risk analysis (cumulative incidence), taking into account death as a competitor, was carried out in order to calculate time to acute myeloid leukemia. All statistical analyses were carried out with RStudio 2022.07.1 (554) and SPSS Statistics version 23 software.

## 3. Results

### 3.1. General Characteristics

A total of 175 patients were included in the study (PMF, *n* = 81 and SMF, *n* = 94). SMF corresponded to post-ET MF (*n* = 53) and post-PV MF (*n* = 41). In SMF, the median elapsed time from MPN diagnosis to MF was 13.3 years (range, 2–33 years) for PV and 10.4 years (range, 3–32 years) for ET. The median age at MF diagnosis was 66 years (range 24–93) and 85 cases (49%) were males. Disease genotype defined by driver mutations evaluated by conventional techniques (PCR or Sanger sequencing) was as follows: MF with *JAK2* mutation (*n* = 113, 65%), MF with *CALR* mutation (*n* = 39, 22%), MF with *MPL* mutation (*n* = 8, 5%) and triple-negative MF (*n* = 15, 9%). G-banding karyotype was available in 74 patients, and a normal karyotype was detected in 34 cases. Main cytogenetic alterations included 20q deletion (*n* = 12), 5q deletion (*n* = 4) and 7q deletion. A complex karyotype was seen in 3 cases and 1 case presented with 17p deletion, affecting *TP53*.

### 3.2. NGS Studies

*JAK2* V617F mutation was detected in 114 patients. Additionally, 4 variants with conflicting interpretations were found in *JAK2* in 3 different patients: two variants in one case (T875N and F694S), and a single variant in the other 2 cases (R1063H and N1108S). Variant T875N was considered likely pathogenic and therefore used for genomic classification [17,18]; variants F694S and N1108S were classified as of uncertain significance and R1063H aslikely benign.

Thirty-nine patients presented *CALR* mutations: type 1 (*n* = 22), type 2 (*n* = 15) and atypical mutations (*n* = 2). A high *CALR* VAF (≥50%) was more frequently observed in patients with type 2 *CALR* mutations (9% versus 47% in type 1 and type 2, respectively, *p* = 0.02). Additional mutations in chromatin/spliceosome genes were observed in 20 cases (51%) whereas 3 cases (8%) presented *TP53* disruption/aneuploidy. The proportion of patients with high *CALR* VAF was similar among patients with and without additional mutations in chromatin/spliceosome genes.

*MPL* variants were detected in 15 patients, including 8 patients with pathogenic variant W515, 2 cases with likely pathogenic mutation S204P, and 5 patients with missense variants of undetermined significance in other positions: L465V (*n* = 1), R592Q (*n* = 2) and Y591D (*n* = 2). Five cases presented with *JAK2* and *MPL* co-mutation, four of them presented *JAK2* V617F with a non-canonical *MPL* variant and one case showed co-mutation of *MPL* W515L and *JAK2* V617F with VAF 83%, and 2%, respectively. Another case presented with co-mutation *MPL* L465V and *CALR* type 2 with 2% and 95% VAF, respectively. Additional mutations in chromatin/spliceosome genes were observed in 8 cases (53%). Among the 10 patients with *MPL* mutations located at W515/S204 positions, 6 cases (60%) presented additional mutations in chromatin/spliceosome genes.

In addition to *JAK2/CALR/MPL*, the most frequent mutated genes were *ASXL1* (*n* = 41, 23%), *TET2* (*n* = 35, 20%), *SRSF2* (*n* = 19, 11%), *U2AF1* (*n* = 19, 11%), *DNMT3A* (*n* = 13, 7%), *TP53* (*n* = 12, 7%), *EZH2* (*n* = 12, 7%), *SF3B1* (*n* = 10, 6%), *ZRSR2* (*n* = 9, 5%), *CBL* (*n* = 7, 4%), *KRAS* (*n* = 6, 3%), *SETBP1* (*n* = 6, 3%), *RUNX1* (*n* = 4, 2%), *PTPN11* (*n* = 4, 2%), *IDH2* (*n* = 3, 2%), *IDH1* (*n* = 2, 1%) and *NRAS* (*n* = 2, 1%). *SRSF2* (*p* = 0.03), *U2AF1* (*p* = 0.05), and *CBL* (*p* = 0.05) were more frequently mutated in PMF compared to SMF. Mutations in splicing genes (*SRFS2*, *U2AF1*, *SF3B1,* and *ZRSR2*) were observed in 43% and 19% of patients with PMF and SMF, respectively (*p* = 0.001). Mutations in the RAS pathway (*KRAS*, *NRAS*, *CBL*) were also more frequent in PMF than in SMF (14% versus 4%, *p* = 0.03). Figure 1 shows which genes had pathogenic or likely pathogenic mutations in each case and the number of mutations per case, according to the type of MF.

### 3.3. Genomic Classification

Genomic classification according to the algorithm described by Grinfeld was as follows: MF with *TP53* disruption (*n* = 16, 9%), MF with a mutation in chromatin/spliceosome genes (*n* = 72, 41%), MF with homozygous *JAK2* mutation (*n* = 37, 21%), MF with heterozygous *JAK2* mutation (*n* = 22, 13%), MF with *CALR* mutation (*n* = 19, 11%), MF with *MPL* mutation (*n* = 4, 2%), MF with other mutation (*n* = 2, 1%) and MF with no known mutation (*n* = 3, 2%). 

Triple-negative MF cases by conventional molecular techniques (*n* = 15) were reclassified by NGS as MF with *TP53* disruption (*n* = 2, 13%), MF with chromatin/spliceosome mutation (*n* = 8, 53%), MF with other mutation (*n* = 1, 7%), MF with no known mutation (*n* = 3, 20%) and other diagnoses (*n* = 1, 7%). Interestingly, a previously diagnosed PV with *JAK2* V617F showed the expansion of a clone with *TP53* mutation with unmutated *JAK2* at the moment of MF progression. *JAK2* V617F negativity at the moment of disease progression was confirmed by digital PCR. The patient with MF with other mutations presented a nonsense variant in *TET2* (R1516*, VAF 32%). The case that finally fulfilled the criteria for an alternative diagnosis showed an aberrant karyotype with t(8;9) and the fusion gene *PCM1::JAK2* was confirmed by NGS; therefore, it was finally reclassified as a myeloid/lymphoid neoplasm with *JAK2* rearrangement, with a clinical picture and bone marrow biopsy compatible with MPN.

The distribution of genomic categories according to MF subgroups is shown in Figure 2. MF with chromatin/spliceosome mutation was more frequent in PMF whereas MF with homozygous *JAK2* mutation and MF with *CALR* mutation prevailed in post-PV and post-ET MF, respectively. The main clinical and hematological characteristics of genomic categories are shown in Table 1. 

MF with *TP53* disruption/aneuploidy was more frequent in SMF than in PMF (12% versus 6%), although the difference was not significant. In 12 patients, mutations in *TP53* were detected by NGS whereas the remaining 4 patients were allocated in this category due to the presence of 5q deletion in G-banding karyotype. Three of these patients presented additional cytogenetic alterations including 7q deletion, 17p deletion, and a complex karyotype in one case each. Those cases with *TP53* mutation, VAF was significantly higher in SMF than in PMF (57% versus 14%, *p* = 0.01). 

MF with mutations in chromatin/spliceosome genes represented the most frequent genomic group in PMF and in post-ET MF accounting for 53% and 36% of the patients, respectively, whereas in post-PV MF the frequency was 24%. Development of transfusion-dependent anemia was observed in 58% of these patients, being more frequent among those receiving JAK inhibitors. In multivariate analysis, patients with chromatin/spliceosome mutations were more likely to develop transfusion-dependent anemia (HR 2.9, 95%CI 1.7–5.0, *p* < 0.0001) adjusted by type of MF (HR 1.1, 95%CI 0.7–1.9, *p* = 0.6) and therapy with JAK inhibitors (HR 2.1, 95%CI 1.2–3.7, *p* = 0.01). 

MF with homozygous *JAK2* mutation (*n* = 40, 23%) was observed in 54% of patients with post-PV MF. It was associated with advanced age and long-standing PV (median elapsed time from PV diagnosis to MF progression, 14.6 years). MF with heterozygous *JAK2* mutation (*n* = 22, 13%) was similarly distributed among PMF and SMF. MF with *CALR* mutation (*n* = 19, 11%) was equally represented by type 1 (*n* = 10) and type 2 (*n* = 9) *CALR* mutations and predominated among post-ET MF. There was a tendency for a higher proportion of high *CALR* VAF (≥50%) in patients carrying type 2 mutations (56% versus 10% or type 2 and type 1 mutation, respectively, *p* = 0.06). A minority of cases belonged to other genomic categories: MF with *MPL* mutation (*n* = 4), MF with other mutation (*n* = 2), and MF with no known mutation (*n* = 3). 

*SF3B1* mutations, not included as a definitory aberration in the chromatin/spliceosome category, were detected in 10 patients (PMF *n* = 5, post-ET MF *n* = 3, post-PV MF *n* = 2). Three of these cases were allocated in the chromatin/spliceosome group due to the presence of concomitant mutations in one or more of the genes defining this category whereas the remaining cases were classified as MF with *CALR* mutation (*n* = 3), MF with *JAK2* heterozygous mutation (*n* = 3) and MF with homozygous *JAK2* mutation (*n* = 1). Five out of these 10 patients with *SF3B1* mutation (50%) developed transfusion-dependent anemia, median survival was 3.9 years, and a 5-year probability of acute myeloid leukemia (AML) of 14%. 

The distribution of genomic subgroups according to risk stratification by IPSS for PMF and MYSEC for SMF is shown in Appendix A. MF with *CALR* mutation, MF with *MPL* mutation, MF with heterozygous *JAK2* mutation, and MF with other mutations were classified predominantly in lower risk categories but the differences were not significant (Appendix A). 

### 3.4. Survival and Progression to Acute Myeloid Leukemia

With a mean follow-up of 6 years, 92 patients have died resulting in a median survival of 7.6 years. There were no significant differences in survival according to MF type (median survival 8.8, 7.3, and 7.9 years in PMF, post-PV MF, and post-ET MF, respectively). Survival was significantly different according to molecular classification (Figure 3). Median survival was 3.5, 6.2, 7.3, and 10.7 years, and not reached for MF with *TP53* disruption/aneuploidy, MF with chromatin/spliceosome mutations, MF with homozygous *JAK2* mutation, MF with *CALR* mutation, and MF with heterozygous *JAK2* mutation, respectively. Molecular high-risk categories, including patients with *TP53* disruption, chromatin/spliceosome mutations, and homozygous *JAK2* mutation, showed a higher risk of death (HR 2.6, 95%CI 1.4–4.9, *p* = 0.003) after correction by type of MF (HR 0.9, 95%CI 0.7–1.2, *p* = 0.4) and IPSS (HR 1.8, 95%CI 1.4–2.4, *p* < 0.001) (see Appendix A). When taking into account only cases with PMF, a higher risk of death was also detected in molecular high-risk categories (HR 2.8, 95%CI 1.1–7.2, *p* = 0.03) and cases with a higher IPSS (HR: 2.1, 95%CI 1.5–2.9, *p* < 0.0001). In an analysis restricted to 39 cases of MF with *CALR* mutation, median survival was 9.6 years (7.9 years and 10.7 years in those with and without chromatin/spliceosome mutation, respectively, *p* = 0.28). High *CALR* VAF was associated with a tendency towards shorter survival (7.1 years and 15.6 years in patients with high and low *CALR* VAF, respectively, *p* = 0.06). In multivariate analysis, high *CALR* VAF was associated with a higher risk of death (HR 2.9, 95%CI 1.04–8.3, *p* = 0.04), after adjusting by the presence of mutations in chromatin/spliceosome genes (HR 1.9, 95%CI 0.75–4.9, *p* = 0.2).

Twenty cases progressed to AML and no differences were observed in the probability of AML according to MF type. These 20 cases corresponded to the following genomic categories: *TP53* disruption/aneuploidy (*n* = 6), chromatin/spliceosome mutations (*n* = 11), and homozygous *JAK2* mutation (*n* = 3), with no case of AML progression recorded in the remaining genomic categories. Five-year probability of AML was 49%, 16%, and 13% for MF with *TP53*, chromatin/spliceosome mutation, and *JAK2* homozygous mutation, respectively (*p* = 0.001 for the comparison according to molecular classification, Figure 4). Those with *TP53* disruption/aneuploidy showed a higher risk of AML (HR 5.7, 95%CI 1.8–18.4, *p* = 0.004) corrected by IPSS (HR 2.0, 95%CI 1.1–3.5, *p* = 0.01) (see Appendix A). 

## 4. Discussion

In the present work, we have analyzed the molecular classification proposed by Grinfeld et al. [10] in a large series of patients with MF from a single institution. To the best of our knowledge, this is the first study aimed to apply this classification taking into account the type of MF. In such genomic classification, 8 genomic categories with different clinical characteristics and outcomes were established after analyzing 2035 MPN patients, 309 of them with MF. In our series MF with chromatin/spliceosome mutations was the most frequent genomic subgroup, accounting for 41% of the patients followed by homozygous *JAK2* mutation in 21%, whereas heterozygous *JAK2* mutation and *CALR* mutation represented the third and fourth group being observed in 13% and 11% of cases, respectively. These figures are superimposable to that reported by Grinfeld et al., except for a lower frequency of heterozygous *JAK2* mutation in our series. Moreover, genomic subgroups including MF with *MPL* mutation, MF with other mutation, or MF with no known mutation were infrequent, representing only 5% of MF patients. 

Genomic classification was significantly different according to the type of MF, confirming different molecular pathways for disease origin and progression in PMF and SMF. The conception of PMF and SMF as different biological entities was particularly illustrative in post-PV MF, where the presence of the homozygous *JAK2* mutation was the hallmark of the disease in most cases. Moreover, the higher frequency of *TP53* disruption/aneuploidy in SMF in our study, a genomic subgroup associated with shorter survival and progression to AML, further supports the separation of these two entities. These findings are in concordance with a recent publication in which a different impact of *ASXL1* mutations was found in PMF and SMF [19].

Mutations in chromatin/spliceosome genes were present in 53%, and 36% of PMF and post-ET MF, respectively. Clinically, this molecular subgroup was characterized by the development of transfusion-dependent anemia and was associated with shorter survival and AML progression. Of note, the inclusion of mutations affecting the RAS pathway (*KRAS, NRAS, CBL*) in this subgroup increases the number of high-risk mutations to take into consideration in MF since the majority of NGS-based prognostic risk models restrict high-risk mutations to *ASXL1/U2AF1/SRSF2/EZH2/IDH1/IDH2* genes [20,21,22]. However, the Grinfeld algorithm and a recent French study including 479 MF patients, both applying Bayesian analysis, identified mutations in the RAS pathway as an adverse prognostic factor [10,23]. Moreover, Coltro et al. reported mutations in the RAS pathway in 13% of patients with MF, being associated with adverse phenotypic features, lower response to JAK inhibitors, and lower survival, especially in PMF [24]. In addition, Loscoco et al. reported *SF3B1* mutations in 7% and 5% of PMF and SMF, respectively, being associated with reduced survival in SMF [25]. We observed *SF3B1* mutations in 10 cases (6%), with these patients showing a higher rate of transfusion-dependent anemia and reduced survival, similar to that observed in patients included in the chromatin/spliceosome category. Our results and others strongly support the inclusion of *SF3B1* mutations as definitory of the chromatin/spliceosome group. 

It has been observed that MF with *CALR* type-1 mutation showed longer survival than those with type-2 mutation [26,27]. In addition, shorter survival and anemia were associated with higher *CALR* VAF in a study including 121 patients with *CALR*-mutated MF, suggesting a more advanced disease [28]. However, high VAF *CALR*-mutated patients were significantly enriched with *ASXL1* mutations indicating that patients with high *CALR* VAF mostly corresponded to MF with chromatin/spliceosome mutations. In contrast, in our series a high *CALR* VAF was more frequently observed in type-2 than in type-1 and was not significantly different according to the presence of chromatin/spliceosome mutations. Moreover, *CALR*-high patients showed an increased risk of death corrected by co-mutations in chromatin/spliceosome genes. These results suggest that homozygosity of the *CALR* mutation might be prognostically relevant explaining in part the shorter survival reported in *CALR* type-2 patients [26,27]. However, the limited number of patients in our study precludes a firm conclusion in this regard.

*JAK2*-mutated patients showed clear clinical differences according to VAF. Outcomes in MF patients with homozygous *JAK2* mutation were similar to those classified in the chromatin/spliceosome group in terms of survival and AML progression, supporting the inclusion of these patients in the molecular high-risk MF group. In contrast, MF with heterozygous *JAK2* mutation showed the longest survival among the different genomic categories and no case progressed to AML, indicating that these cases might be considered as low-risk MF. Finally, very few patients were classified as MF with *MPL* mutation since the majority of *MPL*-mutated patients also carried mutations in one or more genes included in the chromatin/spliceosome group. Similarly, MF with other driver mutations and MF with no known mutation were uncommon representing less than 5% of the patients. 

The main limitations of the present study include its retrospective design and a limited number of patients, which makes it difficult to convincingly interpret the meaning of the molecular classification taking into account the type of MF. Another limitation is that cytogenetic information was not available in a significant proportion of patients and that very unfrequently mutated genes included in the genomic category were not included in our targeted NGS.

This would require collaborative efforts in order to have a high number of patients in each of the 8 molecular groups for each type of MF. However, this work is a good representation of a reference center with a special dedication to the study and treatment of MF.

## 5. Conclusions

Genomic classification confirmed different molecular pathways for disease origin and progression between PMF and SMF. Genomic classification provides relevant information in relation to disease phenotype and outcomes, including progression to acute leukemia. This can be helpful for patient management and therapeutic decisions.

## Figures and Tables

**Figure 1 cancers-15-03904-f001:**
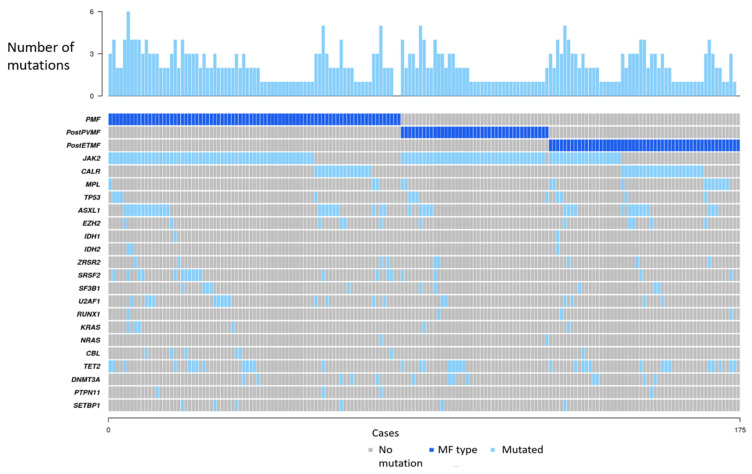
Targeted genes with pathogenic or likely pathogenic mutations in each case, according to the diagnosis of PMF, post-PV MF, or post-ET MF. At the top of the figure, the number of mutations in each case is presented, ranging from no mutation to 6 pathogenic/likely pathogenic mutations. Abbreviations: PMF: primary myelofibrosis. PostPVMF: post-polycythemia vera myelofibrosis. PostETMF: post essential thrombocythemia myelofibrosis.

**Figure 2 cancers-15-03904-f002:**
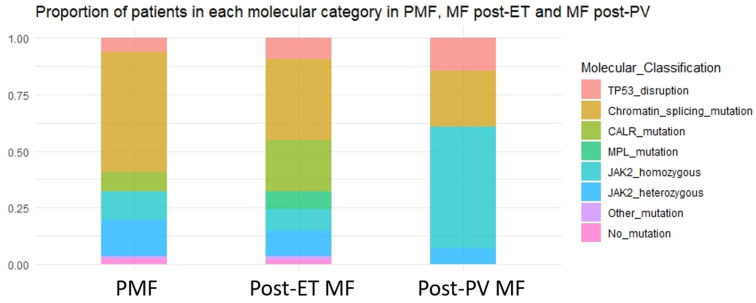
Distribution of patients in each molecular category according to myelofibrosis type: PMF: *TP53* disruption/aneuploidy 6%, chromatin/spliceosome mutations 53%, *CALR* 9%, *MPL* 0%, homozygous *JAK2* 12%, heterozygous *JAK2* 16%, other mutation 1%, no mutation 2.5%. Post-ET MF: *TP53* disruption/aneuploidy 9%, chromatin/spliceosome mutations 36%, *CALR* 23%, *MPL* 7.5%, homozygous *JAK2* 9%, heterozygous *JAK2* 11%, other mutation 2%, no mutation 2%. Groups were significantly different (*p* < 0.0001). Post-PV MF: *TP53* disruption/aneuploidy 15%, chromatin/spliceosome mutations 24%, *CALR* 0%, *MPL* 0%, homozygous *JAK2* 54%, heterozygous *JAK2* 7%, other mutation 0%, no mutation 0%. Abbreviations: PMF: primary myelofibrosis. Post-PV MF: post-polycythemia vera myelofibrosis. Post-ET MF: post essential thrombocythemia myelofibrosis.

**Figure 3 cancers-15-03904-f003:**
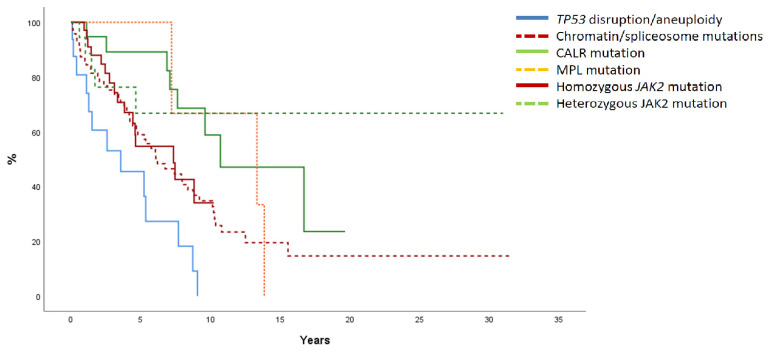
Overall survival in 175 patients with myelofibrosis (MF) according to genomic classification. Kaplan-Meier estimates according to genomic classification. MF with *TP53* disruption/aneuploidy (solid blue line): median survival 3.5 years. MF with chromatin/spliceosome mutations (dotted red line): median survival 6.1 years. MF with homozygous *JAK2* mutation (solid red line): median survival 7.3 years. MF with *CALR* mutation (solid green line): median survival 10.7 years. MF with *MPL* mutation (dotted orange line): median survival 13.3 years. MF with heterozygous *JAK2* mutation (dotted green line): median survival not reached. *p* < 0.0001.

**Figure 4 cancers-15-03904-f004:**
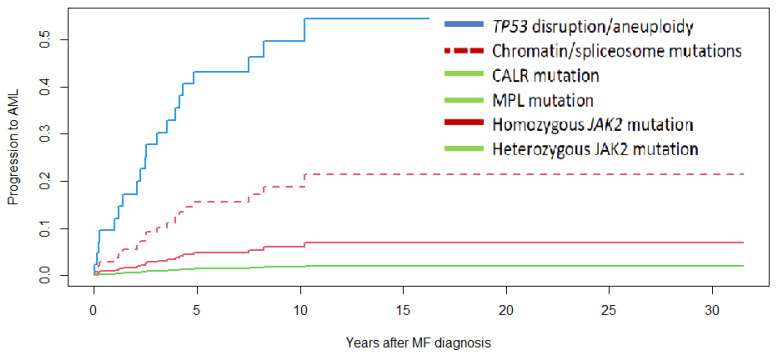
Time to acute myeloid leukemia in 175 patients with myelofibrosis (MF) according to genomic classification. A Cox regression model with a competing risk analysis, taking into account death as a competitor of progression to acute myeloid leukemia has been carried out. MF with *TP53* disruption/aneuploidy (solid blue line). MF with chromatin/spliceosome mutations (dotted red line). MF with homozygous *JAK2* mutation (solid red line). MF with *CALR* mutation, MF with *MPL* mutation, and MF with heterozygous *JAK2* mutation (solid green line). *p* < 0.0001.

**Table 1 cancers-15-03904-t001:** Main clinical characteristics at diagnosis and outcomes in 175 patients with myelofibrosis according to genomic categorization.

	Global *n* = 175	*TP53* Disruption *n* = 16	Chromatin/Splicing Mutation *n* = 72	*CALR* Mutation *n* = 19	*MPL* Mutation *n* = 4	*JAK2* Homozygous Mutation *n* = 37	*JAK2* Heterozygous Mutation *n* = 22	Other Mutation *n* = 2	No Mutation *n* = 3	*p* Value ¶
Age *	66 (24–93)	69 (42–88)	66 (31–89)	57 (24–86)	52 (39–62)	70 (35–88)	66 (25–93)	43 (43–44)	62 (48–78)	0.009
Male sex **	85 (49)	10 (62.5)	45 (62.5)	6 (32)	2 (50)	14 (38)	6 (27)	1 (50)	1 (33)	0.04
Type of MF ** Primary Post-ET Post-PV	81 (46) 53 (30) 41 (24)	5 (31) 5 (31) 6 (37.5)	43 (60) 19 (26) 10 (14)	7 (37) 12 (63) -	- 4 (100) -	10 (27) 5 (13.5) 22 (59.5)	13 (59) 6 (27) 3 (14)	1 (50) 1 (50) -	2 (66) 1 (33) -	<0.0001
Symptomatic splenomegaly **	33 (19)	1 (6)	15 (21)	2 (10.5)	-	8 (22)	6 (27)	1 (50)	-	0.06
Constitutional symptoms **	62 (35)	6 (37.5)	32 (44)	2 (10.5)	-	16 (43)	4 (18)	1 (50)	1 (33)	0.05
Hb, g/L *	106 (63–174)	102 (77–174)	104 (63–150)	102 (84–140)	105 (93–116)	116 (80–159)	108 (87–148)	120 (107–134)	90 (78–92)	0.02
WBC, ×10^9^/L *	10.3 (2–148)	8.6 (2–41)	10 (2.2–133)	6.4 (2.5–26)	8.8 (5.3–15.4)	15.8 (2–148)	8 (4–23)	7.2 (5.2–9.2)	4.5 (4.1–4.6)	ns
Platelets, ×10^9^/L *	307 (18–1640)	157 (18–1640)	269 (35–1086)	467 (78–1359)	254 (112–684)	316 (41–849)	402 (64–1200)	248 (81–416)	255 (27–309)	0.04
Transfusion dependence **	66 (38)	6 (37.5)	42 (58)	5 (26)	3 (75)	7 (20)	2 (9)	1 (50)	0 (0)	<0.0001
JAKi **	109 (62)	9 (56)	46 (64)	16 (84)	2 (50)	22 (59.5)	11 (50)	-	3 (100)	ns
SRV at 5 years	63%	45%	59%	89%	100%	55%	67%	100%	100%	<0.0001
AML at 5 years	14%	49%	16%	0%	0%	13%	0%	0%	0%	0.001

* Median (range), ** *n* (%), ¶ *p* value corresponded to Chi-square, ANOVA, and Log-rank for categorical variables, continuous variables, and K-M estimates, respectively. Abbreviations: MF: myelofibrosis. ET: essential thrombocythemia. PV: polycythemia vera. Hb: Hemoglobin. WBC: white blood cells. JAKi: treatment with JAK inhibitors. SRV: survival. AML: progression to acute myeloid leukemia.

## Data Availability

Data that has been generated during this research project is available from the corresponding author upon reasonable requests. Additional information can be found in Supplemental data files.

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
