# Peer review of "Clinical Characteristics and Outcomes of Patients with Primary and Secondary Myelofibrosis According to the Genomic Classification Using Targeted Next-Generation Sequencing"

_cancers, 2023, doi:10.3390/cancers15153904_

Round 1

Reviewer 1 Report

In this study, authors evaluated the genomic classification which has been proposed by Grinfeld et al. in a large series of MF patients from a single institution, and found distinctive molecular landscapes and prognostic impact of this genomic profiling. Although this study can add the information regarding clinical implication of this genomic classification, there are major concerns that should be addressed by the authors.

- In this study, DNA samples from peripheral blood at diagnosis (75%) or during follow-up (25%) were used. In page 7, line 239, the distribution of genomic subgroups was shown according to risk stratification by IPSS for PMF and MYSEC for SMF. Considering IPSS is developed for the time of diagnosis and DIPSS for anytime of follow-up, how is the distribution according to DIPSS risk stratification?

- In ‘3.4 survival and progression to acute myeloid leukemia’ session, the authors describe the HR(95%CI). The supplementary table for multivariate analysis (factors for OS and/or progression to AML) need to be added.

- For Figure 4, progression curve has to be plotted according to cumulative incidence estimates (death is a competing risk).

- In table 1, if the column of total is added, it is easier to understand the patients’ characteristics.

Author Response

Point 1: In this study, DNA samples from peripheral blood at diagnosis (75%) or during follow-up (25%) were used. In page 7, line 239, the distribution of genomic subgroups was shown according to risk stratification by IPSS for PMF and MYSEC for SMF. Considering IPSS is developed for the time of diagnosis and DIPSS for anytime of follow-up, how is the distribution according to DIPSS risk stratification?

Response 1: Even though 25% of DNA samples were collected during follow-up, they were collected shortly after diagnosis or at a time in which clinical characteristics have not changed significantly since diagnosis. This delay was because there was no DNA available at diagnosis and we did not want to miss the valuable information of these MF patients. For this reason, we consider that the proper scores to calculate are IPSS and MYSEC as we do not provide genomic information at different points of the disease for each case, but some cases have a delay in DNA sample collection. We have clarified this in the main text (page 2, lines 76-78).

Point 2: In ‘3.4 survival and progression to acute myeloid leukemia’ session, the authors describe the HR (95%CI). The supplementary table for multivariate analysis (factors for OS and/or progression to AML) need to be added.

Response 2: The supplementary table for multivariate analysis is added to Supplemental information and the main text is corrected to clarify it.

Point 3: For Figure 4, progression curve has to be plotted according to cumulative incidence estimates (death is a competing risk).

Response 3: A cumulative incidence plot taking into account death as a competing risk is designed as a substitute for Figure 4.

Point 4: In table 1, if the column of total is added, it is easier to understand the patients’ characteristics.

Response 4: A global column has been added to Table 1.

Reviewer 2 Report

Garrote et al in this paper evaluate the genomic classification of myeloproliferative neoplasms recently proposed by Grinfeld et al. in a series of patients with primary and secondary myelofibrosis from a single institution. The paper is well written, but it evaluates a number of patients lower than that of the previously proposed classification, confirming what was previously observed. An evaluation on a higher number of patients, maintaining the sub-analysis proposed by the authors between primary and secondary myelofibrosis, would have been more interesting.

Comments to the authors:

MAJOR COMMENTS

-       The Material and Methods section need to be improved. The authors said that the use 2 different panels to investigate the molecular profile of patients. More information about the differences of the two panels and on the choice of the type of panel are necessary. In addition, it will be useful to highlight how many patients were investigated with one panel or another one, and the frequency and the impact of mutations in genes evaluated in the extended panel and not in the Sophia panel.

-       It would be helpful to show, perhaps as a supplementary table, all the pathogenic or likely pathogenic mutations, as well as the VAF of each mutation.

MINOR COMMENTS

-       The introduction is well written, but I will speak a little bit more about the importance of a molecular classification of MPN which has been recently published (line 60-61)

-       On line 111 the authors said that the JAK2+ patients were 113, while on line 118 they were 114. Please explain.

-       How the authors explained the similar outcome of patients with chromatin/spliceosome mutations and of patients with JAK2 homozygosis mutation?

-       How the authors explained an higher CALR VAF in patients with CALR type-2 mutations?

-       Considering Supplemental Figure 1, I will suggest to stratify also for primary and secondary myelofibrosis.

Author Response

Point 1: The Material and Methods section need to be improved. The authors said that the use 2 different panels to investigate the molecular profile of patients. More information about the differences of the two panels and on the choice of the type of panel are necessary. In addition, it will be useful to highlight how many patients were investigated with one panel or another one, and the frequency and the impact of mutations in genes evaluated in the extended panel and not in the Sophia panel.

Response 1: Both panels are similar as they cover the most frequently mutated genes in myeloid malignancies and almost all genes needed for genomic classification. The reason why two different panels were used for NGS profiling is because we had severe technical issues with one of them (the less extensive panel with 32 genes), that could not be properly addressed by the commercial company. Only cases that were sequenced with good quality were taken into account and cases that were affected by those technical issues were sequenced again with the other panel. As a feasible alternative we found the 40-gene panel that, in addition, had some extra genes that were not covered by our initial option. No pathogenic/likely pathogenic variants were found with the 40-gene panel in those genes that were not included in the 32-gene panel.

Material and Methods section has been extended with extra information regarding which genes were included in each panel and how variant calling and analysis was carried out in each case. Further information regarding genes covered by each panel and their differences can be found in Supplemental Table 2. Moreover, Supplemental Table 3 offers information regarding pathogenic of likely pathogenic variants found in each case.

Point 2: It would be helpful to show, perhaps as a supplementary table, all the pathogenic or likely pathogenic mutations, as well as the VAF of each mutation.

Response 2: we have added a supplemental table with pathogenic and likely pathogenic mutations in each case, with VAF information (see Supplemental Table 3).

 Point 3: The introduction is well written, but I will speak a little bit more about the importance of a molecular classification of MPN which has been recently published (line 60-61)

Response 3: the final part of the introduction has been modified and information regarding MPN molecular classification and its value has been expanded as it follows:

Recently, a molecular classification of MPN including 8 genomic groups has been proposed as a useful tool for personalizing prognosis and treatment of MPN patients [10].

It is a hierarchical classification based on an algorithm that contains 8 categories; molecular and chromosomal abnormalities associated with worse prognosis prevail in the classification. Cases presenting with TP53 disruption or chromatin/spliceosome mutations are included in category 1 and 2, respectively. Categories 3 to 6 include MPN with driver mutations (CALR, MPL or JAK2) that lack definitory abnormalities within categories 1 and 2. Patients with JAK2 mutations are separated in two categories, according to allele burden (categories 5 and 6). Finally, categories 7 and 8 include cases with no known mutations and MPN with other driver mutation, respectively.

This molecular classification offers relevant prognostic information besides MPN driver mutation and it allows to identify those patients at higher risk of progression, disease complications and death, regardless of disease phenotype. However, it has hardly been implemented in clinical practice. The aim of the present study is to evaluate the genomic classification of MF in a large series of MF patients from a single institution, taking into account whether cases are primary or secondary.

Point 4: On line 111 the authors said that the JAK2+ patients were 113, while on line 118 they were 114. Please explain.

Response 4: On line 111 it specifies: Disease genotype defined by driver mutations evaluated by conventional techniques (PCR or Sanger sequencing) was as it follows: MF with JAK2 mutation (n= 113, 65%). On the contrary, on line 121: JAK2 V617F mutation was detected in 114 patients. This difference is because line 111 refers to JAK2 mutation detection by conventional techniques (as specified within the text, with a sensitivity around 10%). As line 121 refers to JAK2 V617F mutations detected by NGS, it means that one case with JAK2 V617F mutation was detected by NGS and not by other techniques due to a low allele burden.

Point 5: How the authors explained the similar outcome of patients with chromatin/spliceosome mutations and of patients with JAK2 homozygosis mutation?

Response 5: In myeloproliferative neoplasms, both chromatin/spliceosome mutations and JAK2 homozygosity are mechanisms that have been associated to disease progression in different circumstances. In our work, both abnormalities can be considered as “high-risk” molecular profiles but are usually found in patients with different clinical circumstances.

MF with chromatin/spliceosome mutation is more commonly found in primary MF patients, with a driver mutation that usually has an allele frequency lower than 50%. In these cases, chromatin/spliceosome mutations probably contribute to a MF clinical picture when associated to the MPN driver mutation: a more aggressive clinical picture, with an initial MF instead of PV or ET. As mutations in these genes, at the end, affect gene expression and DNA transcription, they may contribute to a more ineffective hematopoiesis and progression to AML, more similar to a myelodysplastic syndrome.

On the other hand, MF with homozygous JAK2 mutation is generally found in patients with long-standing PV that develop MF after many years of disease. The mechanism of disease progression probably is a progressive increase in JAK2 allele burden until the non-mutated gene has disappeared in the great majority of MPN clone. Homozygous JAK2 mutated MPN clone has a clear survival advantage over the heterozygous clone, as proliferation and survival of these cells is not sensible to regulation. This may lead to genomic instability and propensity to acquire new aberrations (mutations and chromosomal abnormalities) that also facilitates disease progression to AML and shorten survival.

In conclusion, both mechanisms favor disease progression and, although it probably happens in different circumstances, they result in similar outcomes as both mechanisms facilitate progression to acute leukemia. Further studies regarding clonal evolution of MF and serial NGS is needed in order to elucidate how disease progression happens in those patients more precisely.

Point 6: How the authors explained a higher CALR VAF in patients with CALR type-2 mutations?

Response 6: In our series we found 39 MF that presented with CALR mutation (22 cases with type 1 mutation, 15 cases with type 2 mutation and 2 cases with atypical mutations). In spite of the low number of cases, we found that MF cases with type 1 mutations had a significantly lower allele frequency when compared to MF with CALR type 2 mutation. We did not find differences in the number of additional mutations between type 1 and type 2 CALR MF.

A reasonable explanation to this finding it that CALR type 1 mutation is a mutation with higher pathogenicity and it can act as a driver mutation at lower frequencies and patients can develop the full disease picture. In contrast, CALR type 2 might have less pathogenic power and might need of loss of heterozygosity in order to develop the full myelofibrosis picture. The fact that we did not find a higher number of associated mutations in type 1 CALR cases favours this explanation. Other works focused on CALR MPN have found results that support this theory. Further studies focusing in CALR-mutated myelofibrosis will be needed to dilucidated this.

Pan Y, Wang X, Wen S, Liu X, Yang L, Luo J. The different variant allele frequencies of type I/type II mutations and the distinct molecular landscapes in CALR-mutant essential thrombocythaemia and primary myelofibrosis. Hematology. 2022;27(1):902-908. doi:10.1080/16078454.2022.2107888

Point 7: Considering Supplemental Figure 1, I will suggest to stratify also for primary and secondary myelofibrosis.

Response 7: the member of our team that has the updated information regarding risk stratification is currently out of his work place. We can have the Figure ready at the beginning of August. We apologize for this inconvenient.

Reviewer 3 Report

Dear Author,

I have a few questions and uncertainties 

1.      lines 79-85 - A low cut-off level was adopted (1% or more) - on what basis?

2.      there is no information about NGS coverage in the supplementary materials

3.      the descriptions under the figures are too small and illegible

4.      I do not find information on what NGS platform was used for research

Author Response

Point 1: lines 79-85 - A low cut-off level was adopted (1% or more) - on what basis?

Response 1: It is known that driver mutations in MPN can be found at very low allele frequency (VAF) and, unlike traditional techniques such as Sanger sequencing, NGS allows to detect mutations at lower levels. With a mean coverage around 1000x we found that our panels could reach a sensitivity of 1% in order to detect those mutations at low VAF, especially driver mutations in those cases that were categorized initially as “triple negative” MF. All variants found were filtered with a software and revised manually afterwards in order to avoid false positives. This process was carried out more carefully in all those variants that had VAF <5%. All variants that were considered pathogenic or likely pathogenic with a VAF <5% had to be found in areas with a good coverage, and at least 30 reads of the pathogenic variant, that had to be read in both directions of sequencing. In addition, reported variants had to be described previously as pathogenic or likely pathogenic variants.

Point 2: there is no information about NGS coverage in the supplementary materials

Response 2: We have added a supplementary table with comments in which NGS quality and mean coverage standards are shown, in addition to pathogenic and likely pathogenic mutations found in each case with their allele frequency.

Point 3: the descriptions under the figures are too small and illegible

Response 3: descriptions under the figures have been changed to a bigger size to they can be read properly.

Point 4: I do not find information on what NGS platform was used for research

Response 4: In samples sequenced with Sophia Genetics panel, an Illumina platform was used and sequencing was carried out with MiSeq. In samples in which sequencing was performed with Oncomine Myeloid Assay, Ion Torrent platform was used with Ion GeneStudio S5 system. We have added this information in the main text.

Round 2

Reviewer 1 Report

1. Because a cumulative incidence plot taking into account death as a competing risk is designed as a substitute for Figure 4, in materials and methods statistical description also have to be updated.

2. The authors describe that “Molecular high-risk categories, including patients with TP53 disruption, chromatin/spliceosome mutations and homozygous JAK2 mutation, showed a higher risk of death (HR 2.2, 95%CI 1.2-4.2, p=0.01) after correction by type of MF (HR 0.9, 95%CI 0.7-1.2, p=0.4) and IPSS (HR 1.9, 95%CI, 1.5-2.4, p<0.001)”. Are these consistent with Supplemental Table 4?

3. In page 8 line 276, molecular high-risk categories, including patients with TP53 disruption, chromatin/spliceosome mutations and homozygous JAK2 mutation were described. TP53 disruption/aneuploidy (n=6), chromatin/spliceosome mutations (n=11) and homozygous JAK2 mutation (n=3) progressed to AML. In supplemental Table 5, HR (95% CI) of high molecular risk is 5.7 (1.8-18.4). In page 8 line 297, those with TP53 mutations showed a higher risk of AML (HR 5.7, 95%CI 1.8-18.4, p=0.004), which was exactly same to that of high molecular risk. In supplemental table 5, what does mean “high molecular risk”?

Author Response

Point 1. Because a cumulative incidence plot taking into account death as a competing risk is designed as a substitute for Figure 4, in materials and methods statistical description also have to be updated.

Response 1. The statistical description has been modified in order to include the statistical method used.

Point 2. The authors describe that “Molecular high-risk categories, including patients with TP53 disruption, chromatin/spliceosome mutations and homozygous JAK2 mutation, showed a higher risk of death (HR 2.2, 95%CI 1.2-4.2, p=0.01) after correction by type of MF (HR 0.9, 95%CI 0.7-1.2, p=0.4) and IPSS (HR 1.9, 95%CI, 1.5-2.4, p<0.001)”. Are these consistent with Supplemental Table 4?

Response 1. The text has been updated with the corrected data (last analysis with all included patients is the one in Supplemental Table 4). The term “high molecular risk” has been substituted by “molecular high-risk”, which is the term used in the text and it refers to patients with TP53 disruption, chromatin/spliceosome mutations and homozygous JAK2 mutation. A comment in order to clarify this has been added to Supplemental Table 4, as a footnote.

Point 3. In page 8 line 276, molecular high-risk categories, including patients with TP53 disruption, chromatin/spliceosome mutations and homozygous JAK2 mutation were described. TP53 disruption/aneuploidy (n=6), chromatin/spliceosome mutations (n=11) and homozygous JAK2 mutation (n=3) progressed to AML. In supplemental Table 5, HR (95% CI) of high molecular risk is 5.7 (1.8-18.4). In page 8 line 297, those with TP53 mutations showed a higher risk of AML (HR 5.7, 95%CI 1.8-18.4, p=0.004), which was exactly same to that of high molecular risk. In supplemental table 5, what does mean “high molecular risk”?

Response 1. “High molecular risk” in Supplemental Table 5 has been updated for “MF with TP53 disruption/aneuploidy”, which is the correct term that has to be used here. The text has also been corrected and “TP53 mutations” has been substituted for “TP53 disruption/aneuploidy”.

Thank you very much for your comments.

Reviewer 2 Report

The authors answered all the questions and comments I've made

Author Response

Thank you very much for your comments.

Round 3

Reviewer 1 Report

Authors addressed the questions mostly. However, in the statistics session, the authors have to clarify "time to acute myeloid leukemia curve (figure 4)" be plotted by 1-Kaplan-Meier or cumulative incidence with a competing risk? 

Author Response

Figure 4, where time to acute leukemia is shown, has not been calculated and plotted with a traditional survival analysis but with a survival analysis with competing risks. This means that the endpoint in the analysis can be acute myeloid leukemia, death or "censor" (when none of the events - death or acute leukemia- have happened). This is different from a traditional survival analysis, where only two states are taken into account (event and censor). The analysis that we have carried out is equivalent to "cumulative incidence" but we have used the term "analysis with competing risks" as it is the term used in the documents published this year by Terry Therneau, where is stated "the label 'cumulative incidence' is one of the more unfortunate ones in the survival lexicon, since we normally use `incidence' and `hazard' as interchangeable synonyms but the CI is not a cumulative hazard" (Multi-state models and competing risks, page 3, https://cran.r-project.org/web/packages/survival/vignettes/compete.pdf).

In the plot, only progression to AML curves has been plotted because our variable has 4 states (according to molecular classification), and plotting progression to AML and death (8 curves) was not visual because too much information was present in the same plot and it was not well understood. In the last paragraph of "material and methods" we have specified that the cox regression with survival analysis is a "cumulative incidence" equivalent.

The document that we have followed and adapted in order to do our analysis is the following (see competing risks part): https://cran.r-project.org/web/packages/survival/vignettes/survival.pdf

The R code where the variable with competing risks is created ("censor", "AML", "death" is created is the following:

crdata <- MF_progression_to_AML
crdata$etime <- pmin(crdata$TimetoLAM, crdata$SRV)
crdata$event <- ifelse(crdata$LAM==1, 1, 2*crdata$ESTATUS)

crdata$event <- factor(crdata$event, 0:2, c("censor", "AML", "death"))

cfit <- coxph(Surv(etime, event) ~ Graphic_molecular_class,
                id = ID, data = crdata)
print(cfit, digits=1) # narrow the printout a bit

  coxph(formula = Surv(etime, event) ~ Graphic_molecular_class,
        data = crdata, id = ID)

dummy <- expand.grid(Graphic_molecular_class=1:4)
csurv <- survfit(cfit, newdata=dummy)

plot(csurv[,2], xmax=20*12, xscale=1,
     xlab="Years after MF diagnosis", ylab="Progression to AML",
     col=c(4,2,2,3), lty=c(1,2,1,1), lwd=2)

We hope this answer is convincing. Do not hesitate to contact us if you have more inquiries.